# Distinct HAND2/HAND2-AS1 Expression Levels May Fine-Tune Mesenchymal and Epithelial Cell Plasticity of Human Mesenchymal Stem Cells

**DOI:** 10.3390/ijms242216546

**Published:** 2023-11-20

**Authors:** Rachel Vazana-Netzarim, Yishay Elmalem, Shachar Sofer, Hod Bruck, Naama Danino, Udi Sarig

**Affiliations:** 1The Dr. Miriam and Sheldon Adelson School of Medicine, Department of Morphological Sciences and Teratology, Ariel University, Ariel 4070000, Israel; rachel17ezr@gmail.com (R.V.-N.); naamada@ariel.ac.il (N.D.); 2Department of Chemical Engineering, Faculty of Engineering, Ariel University, Ariel 4070000, Israelshachars@ariel.ac.il (S.S.); hodiest@gmail.com (H.B.)

**Keywords:** HAND2, HAND2-AS1, mesenchymal-to-epithelial transition (MET), epithelial-to-mesenchymal transition (EMT), human mesenchymal stem cells (hMSCs), senescence

## Abstract

We previously developed several successful decellularization strategies that yielded porcine cardiac extracellular matrices (pcECMs) exhibiting tissue-specific bioactivity and bioinductive capacity when cultured with various pluripotent and multipotent stem cells. Here, we study the tissue-specific effects of the pcECM on seeded human mesenchymal stem cell (hMSC) phenotypes using reverse transcribed quantitative polymerase chain reaction (RT-qPCR) arrays for cardiovascular related gene expression. We further corroborated interesting findings at the protein level (flow cytometry and immunological stains) as well as bioinformatically using several mRNA sequencing and protein databases of normal and pathologic adult and embryonic (organogenesis stage) tissue expression. We discovered that upon the seeding of hMSCs on the pcECM, they displayed a partial mesenchymal-to-epithelial transition (MET) toward endothelial phenotypes (CD31^+^) and morphologies, which were preceded by an early spike (~Day 3 onward after seeding) in HAND2 expression at both the mRNA and protein levels compared to that in plate controls. The CRISPR-Cas9 knockout (KO) of HAND2 and its associated antisense long non-coding RNA (HAND2-AS1) regulatory region resulted in proliferation arrest, hypertrophy, and senescent-like morphology. Bioinformatic analyses revealed that HAND2 and HAND2-AS1 are highly correlated in expression and are expressed in many different tissue types albeit at distinct yet tightly regulated expression levels. Deviation (downregulation or upregulation) from these basal tissue expression levels is associated with a long list of pathologies. We thus suggest that HAND2 expression levels may possibly fine-tune hMSCs’ plasticity through affecting senescence and mesenchymal-to-epithelial transition states, through yet unknown mechanisms. Targeting this pathway may open up a promising new therapeutic approach for a wide range of diseases, including cancer, degenerative disorders, and aging. Nevertheless, further investigation is required to validate these findings and better understand the molecular players involved, potential inducers and inhibitors of this pathway, and eventually potential therapeutic applications.

## 1. Introduction

Two key processes guiding tissue and organ development both during embryogenesis and in various pathophysiological contexts are the epithelial-to-mesenchymal transition (EMT) and its reverse mesenchymal-to-epithelial transition (MET) [1]. Epithelial cells form stationary barriers delineating tissues and organs [2], create lumen and tube morphologies (e.g., via ‘hollowing’ and ‘cavitation’ [3]), maintain basal–apical cell polarity via basement membrane (BM) anchoring, and ensure strict cell–cell contact through desmosomes, gap, adherens, and tight junctions [4]. Mesenchymal cells, in contrast, are spindle shaped, reside within, and remodel the 3D fibrous extracellular matrix (ECM), and can travel long distances to invade tissues within the body [2,5]. The epithelial-to-mesenchymal spectrum is wide and contains many intermediate phenotypes [6].

Cell state transitioning (both EMT and MET) is crucial for tissue and organ development and is vital for almost every step of embryogenesis, including extraembryonic trophectoderm specification, embryonic gastrulation, body plan patterning and, ultimately, organogenesis (the establishment and diversification of body organs) [5,7,8,9]. For example, neural crest cells (NCCs) delaminate via an EMT from the neural plate border and migrate to distant embryonic destinations [7,10,11] including the pharyngeal arches [12], head and neck mesenchyme [11], second heart field [13], and adrenal gland medulla [14]. Upon reaching their destinations, NCCs undergo at least one more MET to assume parenchymal differentiated cell states and morphologies [9].

EMTs and METs are also of paramount importance for adult tissue homeostasis and regeneration [15], and are involved in numerous pathological conditions including cancer metastasis [16], fibrosis during aging [2], cellular senescence [17,18], and MET-mediated escape from senescence [19]. In essence, most tissue formation and remodeling processes necessitate at least one EMT–MET cycle, making the study of these processes critical not only for developmental biology and tissue engineering purposes but also for a better understanding of pathologies and possible therapy development.

Global, often antagonistic, cellular programs ensure the protection of epithelial–mesenchymal homeostasis—including specific splicing, miRNA regulatory networks and other epigenetic mechanisms, which are still not completely understood (particularly for MET) [1,5]. Direct human organogenesis embryo models, however, are notoriously difficult to generate (also limited by the ‘14-day’ ethical rule) [20], and EMT/MET regulatory network manipulation is, in many cases, lethal. Hence, current EMT/MET knowledge is mostly deduced from animal models (i.e., drosophila, chick, zebrafish, and murine) [9], and by employing human cancer cell spheroids and/or human primary and stem cell-derived models [21,22,23].

One possible downstream common mediator of MET may be heart and neural crest derivatives 2 (HAND2), a type B basic helix–loop–helix (bHLH) transcription factor (TF) [24], and its associated antisense long non-coding RNA (lncRNA; HAND2-AS1). Embryonically, HAND2 induces type 1 (embryonic) MET patterning of NCCs to branching aortic vessels and of mesodermal second heart field (SHF) cells to the right ventricle (RV), parts of the atria, interventricular septum, and outflow tract [25,26,27]. HAND2 murine knockout (KO) is lethal at ED9.5–10.5, yielding RV hypoplasia, single-chambered hearts, and vascular malformations [25,28]. Molecularly, HAND2 is a crucial downstream regulator of the endocardial VEGF-Notch signaling pathway during cardiogenesis and coronary vasculogenesis [27]. HAND2 also instructs type 1 MET in anterior–posterior limb bud digit patterning (initiating a SHH/FGF feedback loop) [29] and induces embryonic mesothelium progenitor formation delineating internal organs and contributing to organogenesis, tissue homeostasis and regeneration, through unclear signaling [30]. HAND2 was, thus, suggested as a “regulator of tissue morphogenesis and patterning through a mechanism independent of direct DNA binding” [29], indicating tissue context specificity during normal development, as further supported by the evolutionarily conserved roles of bHLH TFs [31].

Our lab’s research focuses on the nexus of tissue engineering (TE [32]) and developmental biology (DB) that is often termed developmental engineering (DE) [33,34,35,36]. While traditional TE uses biomaterial scaffolds and signaling molecules to guide the (top–down) cellular production of biological tissue substitutes, DB studies the basic science principles of tissue, organ, and whole organism formation (bottom–up) [37,38]. DE aims, therefore, to identify, study, and mimic nature’s building blocks and key developmental processes to produce in vitro human tissues and organ models [36].

We uncovered the HAND2-HAND2-AS1 genomic locus in the process of conducting the experiment detailed in this manuscript, involving human mesenchymal stem cells (hMSCs) (as model cells) that were seeded on a bioinductive porcine cardiac extracellular matrix (pcECM) scaffold. We believe that our unintended but fortuitous discoveries made are of interest and possibly have wide implications. We, therefore, hope that this publication will provide an impetus for other researchers to help elucidate the enigma involving the HAND2-HAND2-AS1 regulation of MET in tissue development, homeostasis, regeneration, and pathogenesis.

## 2. Results

We began our work with the decellularization of porcine hearts as we previously published [39,40] to obtain slabs of bioinductive pcECM (Figure 1). To demonstrate the preservation of ECM fibers and structure we stained paraffin sections of the native tissue (as control) and decellularized slabs with H&E and MTC reagents. Additional immunohistochemical (IHC) stains were consequently performed for two cardiovascular-lineage-specific representative growth factors (FGF2 and VEGF, respectively), and an epidermal growth factor (EGF) serving as a negative control (Figure 1, as indicated). These stains, and their associated quantifications and statistical analyses confirmed the significant retention of only the cardiovascular growth factors (VEGF and bFGF), which were organized as pockets within the pcECM following decellularization. This finding is in accordance with previously described mechanisms of ECM biological composition and function [41,42,43,44].

Prior to repopulating our pcECM scaffold with hMSCs (commercially purchased from Lonza, Switzerland), we characterized its cell morphology, marker expression profile, and differentiation potency (Figure 2). These analyses demonstrated tissue culture plastic adherence, a characteristic spindle shaped morphology, consensus marker expression (CD73^+^CD90^+^CD105^+^CD31^−^), and multipotency, following standard protocols for adipose, bone, and cartilage-like cell differentiation. Of note was the absence of CD31 expression by the hMSCs (used as an iso-type negative control), which suggested the lack of any prior endothelial commitment by the cells we used. 

At distinct time points (t = 3, 23 and 30 days) during the culture, we isolated the hMSCs (through trypsinization) and obtained their mRNA. To identify possible induction pathways that may be triggered by the pcECM, the mRNA was then subjected to reverse-transcribed quantitative (realtime) polymerase chain reaction (RT-qPCR) arrays (online Appendix A and Appendix A) for cardiac and endothelial-related genes. Analyzing the data, we observed two interesting and sequential phenomena. At the initial time points following seeding, the HAND2 mRNA levels were significantly upregulated (~2400 fold, relative to standard tissue culture plate expression levels; Appendix A). This upregulation was also observed in repeated experiments in terms of the protein level on Day 4 of culture using IHC cross-sectional histological staining specifically for HAND2 (Figure 3). While the mRNA level of HAND2 subsided to lower values at later time points (Appendix A), the HAND2 protein level remained consistent at least until Day 14 of culture (Figure 3). This phenomenon preceded and then was concomitant with the upregulation of endothelial-related genes, including PECAM1 (CD31), for which the hMSCs were originally negative before seeding (Appendix A). Intrigued by these results, we performed a further protein level evaluation of CD31 expression by the seeded hMSCs using FACS and immunofluorescent stains. The relatively low cell viabilities following cell harvesting for FACS analyses (70–80%) may be attributed to the relatively long exposure to trypsin and processing steps, rather than the cell support ability of the pcECM, which was previously documented as supporting hMSC proliferation strongly [46]. Thus, our results revealed a gradual increase in CD31 expression and presence throughout the 30 days of culture as evidenced via both flow cytometry and by immunofluorescent stains (Figure 3). Collectively, these results suggest that the pcECM induced a MET process in the seeded hMSCs, driving their partial commitment toward an endothelial (simple squamous epithelium) phenotype.

Interestingly, HAND2 only has two exons (Chr. 4q34.1, Genecards database [47]) but its promoter region and first exon partially overlap with an associated antisense long noncoding RNA (lncRNA, HAND2-AS1)—implicated in alleviating numerous human pathologies reviewed in [48,49]. We performed bioinformatic analyses of HAND2 and HAND2-AS1 expression data using known human sequencing databases for adult and fetal normal expression as well as pathological expression levels in various cancers. These bioinformatic data correlations and analyses suggested that in normal adult tissues (Figure 4a,b), during fetal development (Figure 4a,b insets) and also during cancer progression (Figure 4c), HAND2 and HAND2-AS1 expression profiles seem to correlate (Figure 4d) and obey similar trends. In adults, high HAND2-HAND2-AS1 expression levels appear either in tissues that contain neural crest lineage cells and that appear very early in embryonic organogenesis and development (such as the adrenal glands and the heart) or in tissues that are associated with pregnancy (i.e., the endometrium, ovary, and placenta). Other renewable tissues that have high HAND2-HAND2-AS1 expression levels during development (e.g., stomach, kidney, and intestine) seem to lower their expression to a more moderate level in adulthood. The HAND-2 and HAND2-AS1 tissue level expression during homeostasis, however, seems to be highly maintained, as protrusion/imbalance at the tissue level is associated with many different cancer types (Figure 4c). Among these cancer types shown in Figure 4 are two major categories—those with the downregulation of HAND2 and HAND2-AS1 expression relative to that of healthy tissue controls (most cancers analyzed) and those with HAND2 and HAND2-AS1 upregulation (specific for pancreatic adenocarcinoma, PAAD, and for pheochromocytoma and paraganglioma, PCPG).

Intrigued by our bioinformatic analyses results, we performed a hMSC knockout (KO) of both the HAND2 first exon and HAND2-AS1 regulatory/promoter genomic region (Figure 5a, indicated by purple rectangle) using CRISPR-Cas9 (Appendix A). Our initial thought was that given the tissue specificity and the small (only two exons) scale of this gene, knocking it down might not be detrimental to hMSCs. Clearly, using such KO hMSCs (HAND2-HAND2-AS1^null^) would enable the testing of the apparent regulation of the HAND2/HAND2-AS1 gene products on a pcECM-induced hMSC MET toward endothelial phenotypes. 

Surprisingly, while gene editing was successful in obtaining double-‘mutant’ (MUT) KO hMSC cells (Figure. 5b, mix MUT lane, Appendix A), the HAND2/HAND2-AS1 gene products seemed to be vital at least at a basal (low) expression level for hMSC proliferation and function. In fact, within the transfected cell populations appearing on the same plate and culture conditions (Figure 5c), two cell morphologies became apparent following CRISPR-Cas9 editing. One rare morphology (three discrete colonies) was identical to the wildtype (WT) hMSC morphology. These cells continued to proliferate and were capable of subculturing, i.e., localized short and low-volume trypsinization enabled the selectively lifting and propagation of them in other tissue culture dishes. Once propagated, these colonies’ clones (marked as clone 1–clone 3, lanes 2–4, Figure 5b) exhibited a heterozygous HAND2^+/−^ PCR product containing both a MUT/KO phenotype of 260 bp (short/edited) and 976 bp (WT) PCR product bands on agarose gel electrophoresis. Most of the cell population, however, displayed a senescent-like hypertrophic and cell cycle arrested morphological phenotype (Figure 5C, MUT marked morphologies compared to WT/colony standard morphology). These cells did not organize into colonies and remained hypertrophic and quiescent/senescent for at least three weeks on the plate. Of note is that given the pressure of puromycin selection, no homozygous HAND2^+/+^ cells were present on the plate. The mRNA of these cell types was harvested (labeled mix MUT in Figure 5b) and we saw that their PCR product displayed a homozygous short HAND2^−/−^ band. This band was further sanger-sequenced, confirming that the actual deletion sequence was correct (as designed, Appendix A). Given the paucity of this cell type and the lack of its proliferation, we were unable to perform additional characterization.

Nevertheless, and taken together with the results of HAND2 involvement in MET induction, these observations further suggest a distinct concentration-dependent HAND2/HAND2-AS1 control of cell phenotype. That is, no HAND2/HAND2-AS1 expression (complete absence) causes senescence (or at minimum cell cycle arrest and cellular hypertrophy). Basal low–intermediate expression is critical to maintain the mesenchymal phenotype, and high expression levels may induce a MET toward epithelial/endothelial phenotypes. To the best of our knowledge, this finding is new, may be of potential interest to the readership of this journal, and merits further validation and elucidation.

## 3. Discussion

The bioactivity of the pcECM scaffold material in vitro [46,52,53,54] and in vivo [41,55] has been thoroughly described by our group and others. The descriptions include the induction of cardiac commitment in human-induced pluripotent stem cells (hiPSCs) in the absence of induction media [53], stimulation of neo-tissue formation [52], and modulation of host immunity resulting in the recruitment of host progenitor cells that assist in cardiac regeneration even after scarring in rat myocardial infarction models [41,55]. This characterization is in accordance with an established body of evidence regarding the biological activity, tissue specificity, and induction capacity of decellularized ECM-based materials [42,43,44]. Thus, in the absence of an immune response (in a completely in vitro setting), the gradual nature of the effect observed may be attributed to dependency on ECM remodeling by the cells, which help release localized high concentrations of growth factors retained within the decellularized composite ECM. Such high localized concentrations may be a strong driving force in selectively directing stem cell fate decisions and differentiation states in a tissue-specific manner, which is dependent on the relative abundance and composition of tissue resident growth factors.

Regardless of the actual activation molecular pathway involved, our findings clearly indicate the early (from days 3–4 and up to at least 14 days) activation of HAND2 in hMSCs seeded on the pcECM, which precedes their MET toward endothelial phenotypes. Our bioinformatic analyses further revealed that HAND2 and HAND2-AS1 expression levels are tightly correlated (Pearson coefficient > 0.90) across many different tissues both during embryonic organogenesis (between weeks 10 and 20), as well as in adult normal and pathologic conditions. This tight regulation suggests that HAND2-HAND2-AS1 may be involved in a centrally conserved and important regulatory mechanism, the nature of which is currently unknown. Our findings suggest that this regulatory pathway results in highly active HAND2 expression during organogenesis. This HAND2 expression is subsequently reduced in most adult tissues to basal medium–low levels, except for the endometrium, heart, adrenal and placenta, where HAND2 expression levels remain high even in adulthood. A common denominator among these high HAND2-expressing tissues may be the fact that they are all critical for embryonic support and pregnancy-related hormonal regulation. In addition, human protein level analyses [56] for HAND2 suggest its constant low–medium level expression, in many tissues, beyond those that were identified via our sequencing data analyses, and through a histologically verified manner. 

Taken together, we suggest that this HAND2-HAND2-AS1 regulatory pathway may represent a fine-tuning mechanism regulating plasticity in a spectrum/axis comprising a senescence-like phenotype (a complete absence of HAND2/HAND2-AS1 expression), the maintenance of mesodermal/mesenchyme stemness/proliferation capacity (at basal expression levels), and MET (at high levels). This suggested mechanism is further supported by the similarity of adult hMSCs to neural crest cells, which are considered the fourth germ layer, the literary evidence suggesting that HAND2/HAND2-AS1 has role as a regulator of many cancer types, and the critical role of MET in cancer progression and metastasis, as outlined below.

At the molecular level, the regulation of adult endometrial HAND2 function is the most substantiated. At this level, it is induced by upstream progesterone (P4)-Indian hedgehog (IHH) signaling during menstrual cycles and initiates a downstream MET (also termed decidualization [57])—a key requirement for blastocyst receptivity during implantation [58]. In wider adult tissue pathological contexts (type 2/3 MET), however, HAND2 is associated with endometriosis induction (upregulating proinflammatory IL15 expression [59,60]), right ventricle protection from pressure overload damage [26], familial dilated cardiomyopathy [61], the onset of mesothelioma malignancies [30], and obesity (regulated by glucocorticoid signaling) [62]. HAND2 is also silenced in colorectal cancer (via hypermethylation correlated with poor prognosis) and has been identified as an epigenetic driver gene and a potential tumor suppressor (via MAPK/ERK signaling) [63]. Interestingly, HAND2 and its associated lncRNA, HAND2-AS1, were implicated in alleviating numerous human pathologies reviewed in [48,49]. It is reasonable to assume that a HAND-AS1-induced MET can be a shared underlying mechanism initiating atherosclerosis prevention [64,65], the impairment of fibroblast activation in rheumatoid arthritis [66], and the inhibition of glioma [67], prostate [68], ovarian [69,70], gastric [71], bladder [72], cervical [73,74], breast [75], and pancreatic [76] cancer progression, with mixed reports in liver cancer [77,78,79].

To date, and to the best of our knowledge, HAND2 function was not effectively studied using in vitro cell and tissue models of MET. One study used transcriptomic analysis of primary human cardiac mesenchymal stromal cells (cMSCs) to identify age-dependent biological pathways regulating immune responses and angiogenesis. In that study some of the high HAND2 expressing hMSC displayed increased capacity for MET toward an endothelial (simple squamous epithelium) state in a senescent selective manner (dependent on CD90+ pre-expression) [18]. Other studies using bone marrow hMSCs showed that IHH signaling affects cellular senescence through the modulation of ROS/mTOR/4EBP1 and p70S6K1/2 signaling—a key molecular pathway in aging [80]. While HAND2 was not directly explored in the latter study, senescence dependency on IHH signaling may also be relevant to HAND2 activation (as occurring in the endometrium). This merits further elucidation using adequate cell and tissue models. 

## 4. Materials and Methods

### 4.1. Decellularization and pcECM Characterization

Acellular pcECM samples (15 × 10 × 1.5 mm) were produced as we previously reported [39,40,52]. Briefly, native porcine left ventricular slices were immersed sequentially in alternating hyper-hypotonic solutions (1.1%*w*/*v* and 0.7%*w*/*v* NaCl, respectively), a trypsin (0.05%*w*/*v*)-EDTA (0.02%*w*/*v*) solution, and a 1%*v*/*v* Triton™-x-100 (in 0.1%*v*/*v* ammonium hydroxide) PBS solution. All decellularization reagents were purchased from Sigma-Aldrich (St. Louis, MO, USA). Prior to experimentation, sliced matrices were disinfected with 70%*v*/*v* ethanol and washed with PBS containing a double-antibiotic–antimycotic concentration (2%*v*/*v*, Gibco™, TermoFisher Scientifc, Waltham, MA, USA) followed by immersion in cell culture media overnight at 37 °C and in 5% CO_2_.

For histological analyses, three representative samples of each experimental group were fixated in fresh paraformaldehyde (4%*w*/*v* solution, PFA, Sigma-Aldrich, St. Lous, MO, USA) for 2 h, and then processed for paraffin blocks. The native tissue of similar dimensions was used as the positive control. Paraffin sections (5 µm) were processed for hematoxylin and eosin (H&E) and Masson’s trichrome (MTC) stains. For immunohistochemical stains, we used routine IHC staining protocols. The H&E, MTC, and IHC protocols used were the same as those we previously published [41]. For IHC stains, we used the following primary antibodies: rabbit IgG-anti epidermal growth factor (EGF, abcam, Cambridge, UK, ab9695), mouse IgG anti-basic fibroblast growth factor (bFGF, Merck-Millipore, Burlington, MA, USA, 05-118), and mouse IgG anti-vascular endothelial growth factor A (VEGF, abcam, ab1316). For all stainings, the primary antibody dilution was 1:100 and antigen retrieval steps were performed in a pressure cooker for 40 min. Secondary antibody staining was performed with the Novolink polymer detection system (RE7290-K, Leica Biosystems, Nußloch, Germany), in accordance with the manufacturer’s instructions.

The quantification of positive growth factor IHC staining was performed using the FIJI open-source image processing package [45] with built-in plugins for the measurement of the relative positively stained area based on a maximal threshold setup of 8-bit converted images. The same threshold was used for both samples and controls. At least n = 4 images were used for each group for quantification purposes. 

### 4.2. hMSC Culture and Characterization

hMSCs were commercially purchased (Poietics^TM^, Lonza, Basel, Switzerland, PT-2501) and guaranteed by the supplier to be CD105^+^CD166^+^CD29^+^CD44^+^CD14^−^CD34^−^CD45^−^ (via flow cytometry) and capable of differentiating down the adipogenic, chondrogenic, and osteogenic lineages when cultured in the recommended differentiation medium [81]. Cells were cultured in standard cell culture dishes using α-MEM with 10%*v*/*v* FBS, 1%*v*/*v* antibiotic–antimycotic solution, and 1%*v*/*v* of a 200 mM L-glutamine solution. bFGF (5 × 10^−3^ µg/mL) was supplemented every other day. Cells were split at 70–80% confluency at a 1:4 ratio and were used up until passage 4–5 in all experiments described herein.

To visualize the cell expression of characteristic markers, we performed immunofluorescent staining on glass cover slides for representative positive and negative markers. Briefly, cells were cultured on round 18 mm (0.2 mm thick) cover slides on a standard tissue culture plate. Upon reaching 70% confluence, cells were fixed for 5 min to the slides using ice cold methanol, followed by partial brief air drying. Next, the slides were blocked with 3%*v*/*v* FBS containing PBS and incubated in a humidity chamber for 30 min at RT with the following mouse anti-human primary antibodies: CD90 (BD Biosciences, Franklin Lakes, NJ, USA, 555593), CD105 (BD Biosciences, 555690), CD73 (Santa Cruz Biotechnology, Inc., Dallas, TX, USA, 32299), and CD31 used as an isotype negative control (R&D Systems, Minneapolis, MN, USA, BBA7) all at a 1:50 dilution in 3%*v*/*v* FBS containing PBS. The subsequent primary antibody wash (2×) slides were incubated with a PE anti-mouse (BD Biosciences 550589, dilution 1:100) secondary antibody for 30 min at RT. The slides, prepared thereafter, were washed in the same serum containing wash buffer and mounted with aqueous mounting medium containing DAPI (DAPI Fluromount-G^®^, SouthernBiotech, Birmingham, AL, USA). Images were acquired using a Nikon Eclipse TE2000-U equipped with a fluorescent camera at 400× magnification (Nikon Corporation, Tokyo, Japan). 

To assess the cells’ differentiation capacity at higher passages (passage 5), we performed cell differentiation using commercial differentiation kits. A cell solution of 1.6 × 10^7^ cells/mL, 5 × 10^3^ cells/mL, and 1 × 10^4^ cells/mL of hMSCs were prepared with complete α-MEM media, for chondrogenesis, osteogenesis, and adipogenesis differentiation, respectively (as recommended by the manufacturer’s protocols). Briefly, 10µL droplets of cells of each cell concentrations were added to the center of each well of 12-well plates and incubated in a standard 37 °C 5% CO_2_ incubator for 2 h. The α-MEM media were then replaced with differentiation media, StemPro^®^ Chondrogenesis Differentiation Kit, StemPro^®^ Osteogenesis Differentiation Kit, and StemPro^®^ Adipogenesis Differentiation Kit (Gibco™, TermoFisher Scientifc, Waltham, MA, USA), and replenished every other day. Staining with respective dye indicators—Alcian Blue for chondrogenesis, Alizarin Red S for osteogenesis and Oil Red O for adipogenesis (all purchased from ThermoFisher Scientific, Waltham, MA, USA)—was performed at the end of the differentiation process. 

### 4.3. pcECM-Induced MET in hMSCs 

The seeding of hMSCs on the pcECM was performed on pre-cut disinfected, and culture media-incubated pcECMs (96-well plate format) at a density of 3 × 10^5^ cells/cm^2^ as previously published [46]. hMSC-seeded pcECM constructs were cultured for 30 days. At designated time points throughout the culturing period, mRNA was extracted (phenol and guanidine thiocyanate, qiasol^TM^, Qiagen, Hilden, Germany) from at least n = 3 samples per time point (t = 4-, 23- and 30 days post-seeding). The integrity of extracted mRNA was evaluated using 2100 Bioanalyzer (Agilent, Santa Clara, CA, USA), and RIN values of >7 were used for subsequent steps. mRNA was reverse-transcribed to cDNA using RT^2^ First Strand Kit (330401, Qiagen) followed by real-time qPCR analyses using two separate Qiagen RT^2^ PCR arrays custom-designed with Qiagen for the specific lineage identification of endothelial and cardiomyocyte cell phenotypes (given the cardiovascular origin of the pcECM). Gene array composition and identities appear in Appendix A.

Qualitative detection of HAND2 and CD31 expression in hMSCs at the protein level in situ (on the pcECM) was performed via immunohistochemistry (IHC) and immunofluorescence (IF), respectively. HAND2 IHC was performed using rabbit IgG anti-human HAND2 (abcepta, San Diego, CA, USA, AP17008C, 1:100) per the protocol described above for pcECM growth factor staining with slight modifications that included antigen retrieval at pH = 9, and 0.4%*v*/*v* Triton-X-100 perforation for intracellular staining. CD31 IF was similarly performed on fixated tissue sections using the R&D systems, BBA7 mouse IgG anti-human CD31 primary antibody diluted to a ratio of 1:50 in 3%*v*/*v* FBS containing PBS, and a secondary PE anti-mouse (BD Biosciences 550589, dilution 1:100) antibody for 30 min RT. Dapi was used for counterstaining. Images were acquired using a fluorescent and brightfield inverted microscope (Nikon eclipse TE-2000U). 

In addition, and as a more quantitative measure that avoids potential background fluorescence from the ECM, hMSCs were also harvested at designated culture time points from the pcECM and subsequently FACS analyzed for the quantification of CD31. For cell harvesting, hMSC-seeded pcECMs were washed twice with 0.625 mM EDTA in PBS and transferred into 0.5 mL of Tryple 10X (ThermoFisher Scientific, Waltham, MA, USA) in a 1.5 mL Eppendorf tube. After an 8 min incubation, pcECMs were removed and 0.5 mL of complete culture media (containing 10%*v*/*v* FBS) was added in each Eppendorf tube before centrifugation at 300 g for 5 min. All cell pellets were collected in a single tube, washed twice with cold (4 °C) PBS, and then immunofluorescence-stained, with a short formaldehyde fixation step of 30 min on ice, against CD31 (Biolegend, San Diego, CA, USA, 303102, at a 1:50 dilution) and PE anti-mouse (BD Biosciences 550589, at a 1:100 dilution) for flow cytometry analysis using a BD FACS Calibur^TM^ machine. Tissue culture plate cells served as a control for this experiment. Representative results are shown out of at least n = 3 samples at each time point and experimental condition as indicated.

### 4.4. Bioinformatic Analyses

Bioinformatic analyses for HAND2 and HAND2-AS1 expression levels were performed using several separate datasets: The National Center for Biotechnology Information (NCBI) bioproject #PRJNA280600 (“RNA sequencing of total RNA from 20 human tissues” [50]); the NCBI bioproject #PRJNA270632 (“Tissue-specific circular RNA induction during human fetal development” [51]); the National Institutes of Health (NIH)- and National Cancer Institute (NCI)-supported Cancer Genome Atlas Program (TCGA, https://www.cancer.gov/tcga, accessed on 18 November 2023); and the Genotype-Tissue Expression (GTEx) project database (dbGaP accession #phs000424.vN.pN [82]) supported by the Common Fund of the Office of the Director of the NIH, and by NCI, NHGRI, NHLBI, NIDA, NIMH, and NINDS. Datasets from the NCBI bioprojects were copied from the NCBI Gene Resource for HAND2 (Gene ID #9464) and HAND2-AS1 (Gene ID #79804). TCGA and GTEx data were obtained through the gene expression profiling interactive analysis 2 (GEPIA2) server [83]. All data obtained were further statistically analyzed and presented using GraphPad Prism version 10.0.2 (232) for Windows (GraphPad Software, Boston, MA, USA, www.graphpad.com, accessed on 18 November 2023).

### 4.5. CRISPR-Cas9 Knockout of HAND2-HAND2-AS1 in hMSCs

In an effort to be as succinct as possible here, the detailed CRISPR-Cas9 protocol and design employed appear in the Appendix A.

### 4.6. Statistical Analyses

Unless otherwise stated, results indicate a mean ± standard deviation of at least n = 3 biological repetitions per sample group and time point. For statistical analyses, results were first tested for normality using the Shapiro–Wilk test criteria. All results obeyed Gaussian normal distribution and were, therefore, analyzed using either a *t*-test or one-way analysis of variance (ANOVA) with Tukey’s honest significant difference (HSD) post hoc correction for multiple comparisons, when appropriate. Adjusted *p* values tested are indicated by (*, *p* < 0.05), (**, *p* < 0.01), (***, *p* < 0.001), and (****, *p* < 0.0001) in the appropriate figures. (ns) indicates a non-significant difference. All statistical tests and graphical results were analyzed and displayed using GraphPad Prism software version 10.0.2. 

## 5. Conclusions

Collectively, HAND2 and HAND2-AS1 are suggested herein as common downstream regulators of the MET during embryonic cardiac tissue development (as well as in digit development), in the adult heart, endometrium, renewable epithelial and adipose tissues, and within wider tissue contexts of senescence and related pathologies. It is also reasonable to assume that the regulation of MET (which results in a wide spectrum of commitment levels) would occur in a manner that is proportional to that of the expression level. Indeed, several different HAND2 activators (cardiac Notch, endometrial progesterone-IHH, and adipose glucocorticoid signaling) along with its lncRNA (HAND2-AS1) epigenetic regulation pinpoint the HAND2-HAND2-AS1 loci as a possible local integrator and driver of MET in tissue development, renewal, and senescence. The fascinating MET roles of HAND2 and HAND2-AS1 within such diverse biological contexts remain to be elucidated and, to this end, necessitate the development of appropriate inflammation-free human 3D cell and tissue models.

Our suggested mechanism has far-reaching implications for developmental biology, regenerative medicine, and tissue engineering, and as a possibly common treatment target for many related pathologies. Nevertheless, further investigations and efforts are required to verify and corroborate our findings and suggested mechanism, to better understand this regulatory pathway, identify a more controlled environment that can selectively induce and inhibit this pathway, and ultimately understand the molecular players involved. We also suggest that targeting this pathway may represent a promising new therapeutic approach for a wide range of diseases, including cancer, degenerative disorders, and aging.

## Figures and Tables

**Figure 1 ijms-24-16546-f001:**
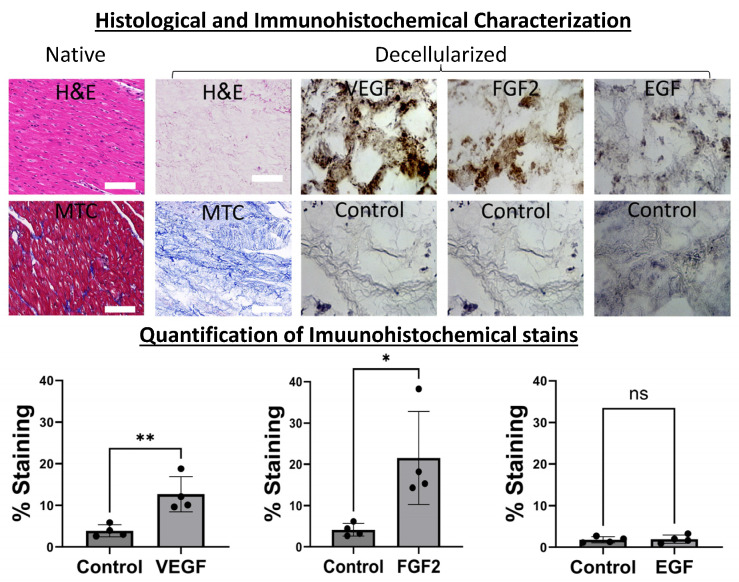
Histological and immunohistochemical characterization of bioactive decellularized pcECM. The pcECM is acellular and preserves cardiovascular-tissue-specific ultrastructure and bioactive growth factors. A decellularized pcECM was paraffin-cross-sectioned and compared to native porcine left ventricular tissue using H&E and MTC stains, as indicated. Sequential paraffin cross-sections of the pcECM were also immunohistochemically (IHC) stained, as indicated, for two representative cardiovascular-related growth factors: basic fibroblast growth factor (bFGF) and vascular endothelial growth factor A (VEGF). IHC staining for epidermal growth factor (EGF) is also presented as a control, along with isotype staining for each case. Scale bars: 100 μm. Microscope magnifications: 200×. Representative images out of at least n = 3 biological samples and n = 2 IHC/histological section stains tested for each group are presented. A quantification of immunohistochemical stains was performed using the FIJI open-source image analysis and processing package [45] in n = 4 regions of interest in each group. Statistical *t*-test comparisons were performed using GraphPad Prism version 10.0.2. (*) *p* < 0.05; (**) *p* < 0.01, (ns) non-significant.

**Figure 2 ijms-24-16546-f002:**
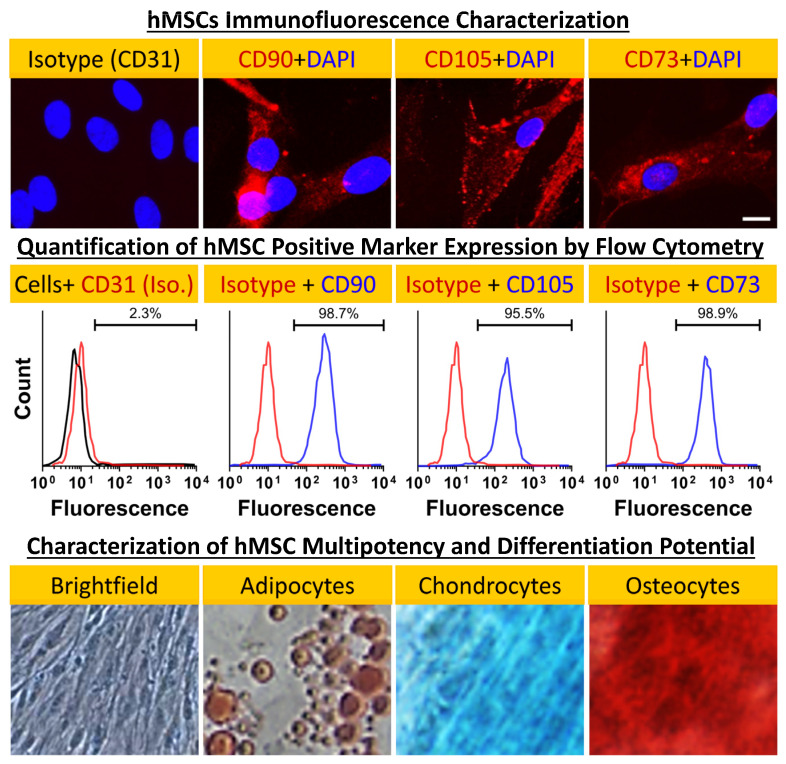
hMSC characterization. Negative (CD31^−^, isotype control) and positive (CD90^+^CD105^+^CD73^+^, as indicated) marker stains (first row) and fluorescent activated cell sorting (FACS) analyses and quantification results (second row) for bone marrow hMSCs (Lonza, Switzerland) showing consensus marker expression. Cellular morphology even at passage 5 remained characteristic of hMSCs as indicated by brightfield phase contrast microscopy. Furthermore, the cells’ differentiation potential appeared to be conserved at that passage as well—shown here for hMSC-derived differentiated adipocyte- (oil red), chondrocyte- (Alcian blue) and osteocyte- (Alizarin red S) like cells. The brightfield image shows the untreated hMSC control’s morphology. Magnifications: brightfield image: 100×; fluorescent images: 200×; differentiation images: 40×.

**Figure 3 ijms-24-16546-f003:**
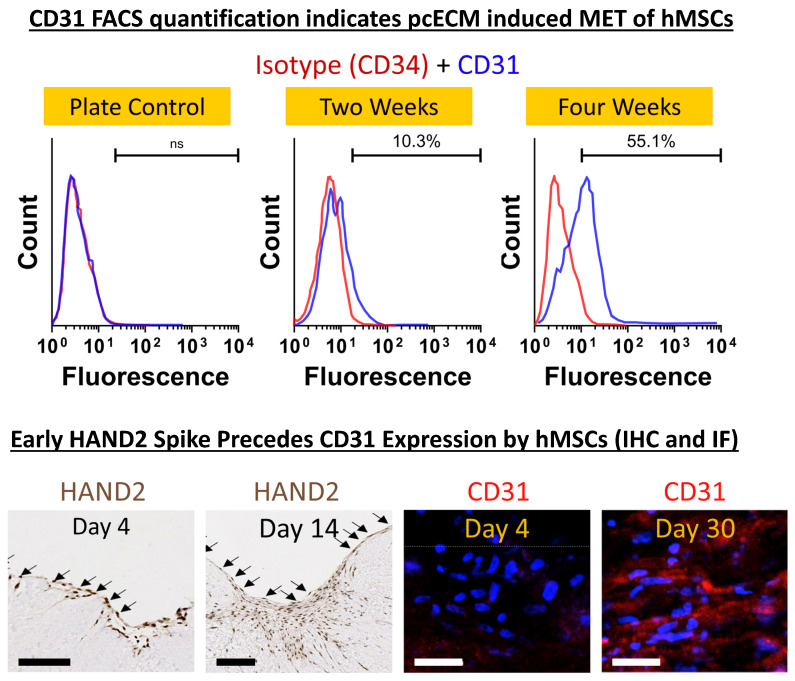
pcECM-induced early HAND2 expression followed by gradual MET toward an endothelial phenotype. The protein level expression of endothelial markers is demonstrated by CD31/PECAM^+^ presence in both trypsinized hMSCs (using FACS, at two and four weeks after seeding; top row) and in situ immunofluorescence imaging (CD31 as indicated). This is in sharp contrast to the lack of CD31 expression by these cell types on the plate (FACS, plate control) or pcECM (Day 4 immunofluorescent stains), as indicated. For FACS analyses, cell viability was assessed via trypan blue stains immediately after trypsinization and was ~70–80%. Three to four days after seeding, hMSCs on the pcECM expressed high levels of HAND2 at the mRNA (Appendix A) and high protein levels (HAND2—Day 4 nucleus localization; IHC, bottom left image as indicated). The HAND2 transcription factor remained co-localized in the hMSC nucleus for at least 14 days on the pcECM as demonstrated via IHC stains. This was accompanied by hMSCs exhibiting an unusual squamous monolayer epithelial-like morphology (arrows in HAND2 IHC images) and was followed by a gradual increase in the expression of endothelial cell markers through long-term (up to 30 days) cultures (Appendix A). **Sample size**: For FACS, n = 3 biological repetitions for each group; for IHC and IF, n = 3 biological samples per measure and at least two sections taken for each stain. **Scale bars**: IHC HAND2 stain: 500 µm. CD31 IF stain: 100 µm.

**Figure 4 ijms-24-16546-f004:**
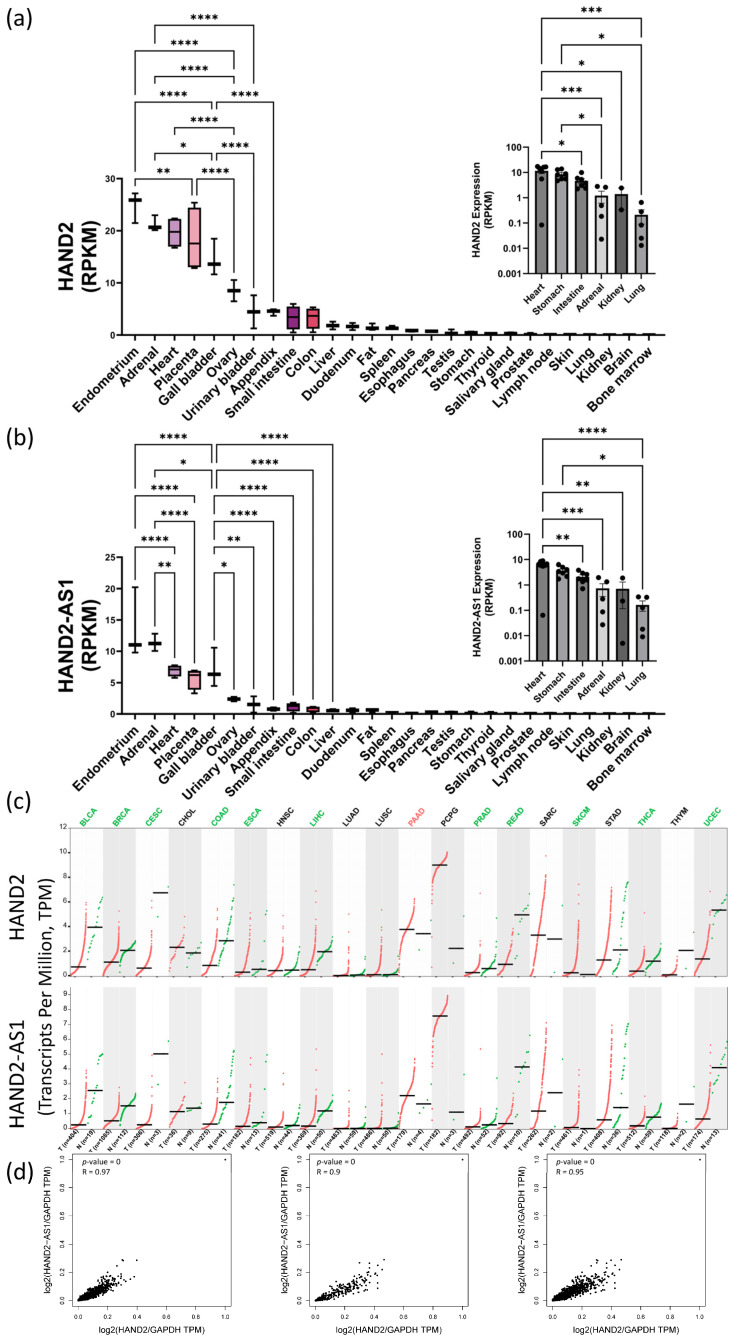
Bioinformatic analyses of HAND2 and its associated lncRNA HAND2-AS1 expression patterns in adult, embryonic, and pathological tissues. Bioinformatic analyses of HAND2 (NCBI Gene accession: 9464) (**a**) and HAND2-AS1 (NCBI Gene accession: 79804) (**b**) expression in adult human tissues (reads per kilobase per million reads, RPKM) based on the National Center for Biotechnology Information (NCBI) bioproject #PRJNA280600 (“RNA sequencing of total RNA from 20 human tissues” [50]). Insets in (**a**,**b**) show corresponding HAND2 and HAND2-AS1 embryonic expression, respectively, in sampled tissues between weeks 10 and 20 of human gestation as sourced from the NCBI bioproject #PRJNA270632 (“Tissue-specific circular RNA induction during human fetal development” [51]). Expression levels of HAND2 and HAND2-AS1 (RPKM, as indicated) in a collection of cancer samples (red dots), relative to neighboring normal tissue samples (green dots), are shown side by side in (**c**) for various cancer types abbreviated as follows: BLCA—bladder urothelial carcinoma; BRCA—breast invasive carcinoma; CESC—cervical squamous cell carcinoma and endocervical adenocarcinoma; CHOL—cholangiocarcinoma; COAD—colon adenocarcinoma; ESCA—esophageal carcinoma; HNSC—head and neck squamous cell carcinoma; LIHC—liver hepatocellular carcinoma; LUAD—lung adenocarcinoma; LUSC—lung squamous cell carcinoma; PAAD—pancreatic adenocarcinoma; PCPG—pheochromocytoma and paraganglioma; PRAD—prostate adenocarcinoma; READ—rectum adenocarcinoma; SARC—sarcoma; SKCM—skin cutaneous melanoma; STAD—stomach adenocarcinoma; THCA—thyroid carcinoma; THYM—thymoma; and UCEC—uterine corpus endometrial carcinoma. Cancer data were obtained from the NIH- and NCI-supported TCGA, and from the GTEx project databases (data sources appear in Section 2). Correlations between HAND2 and HAND2-AS1 expression levels are shown in normal tissues ((**d**), left), tumor tissues ((**d**), middle), and combined normal and tumor tissues ((**d**), right), combining both the TCGA and GTEx data sets. (*) *p* < 0.05; (**) *p* < 0.01, (***) *p* < 0.001, (****) *p* < 0.0001.

**Figure 5 ijms-24-16546-f005:**
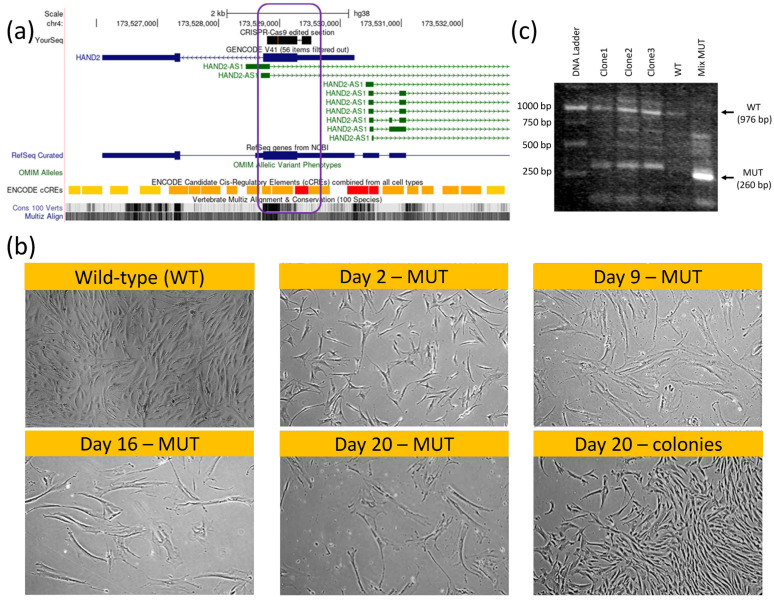
HAND2/HAND2-AS1 locus knockout (KO) hMSCs using CRISPR-Cas9 genomic editing. Genome view of HAND2 (sense strand, blue) and HAND2-AS1 isoforms (antisense strand, green) sharing a common promoter region ((**a**), red ENCODE region), in which the permanent KO ((**b**), CRISPR–Cas9 black sequence top lane; the KO region is graphically displayed by a purple rectangle; the KO vector map is provided in Appendix A) led to a hMSC senescent morphology (**b**); MUT-marked images show time progression of proliferation–arrest and hypertrophy within the same wells, without passaging compared to wild-type (WT) and to heterozygous HAND2^+/−^cells that still managed to form characteristic hMSC colonies ((**b**), as indicated). Non-edited WT cells were eliminated via puromycin selection (Appendix A). Given the scarcity of full MUT cells (non-dividing), we compared HAND2 PCR products using gel electrophoresis (**c**) for the original (WT, untreated) and total gene-edited plates (mixed MUT) comprising mostly HAND2^−/−^ senescent cells. Colony clones (clone 1–clone 3) displayed a mixed WT-MUT band signature, suggesting heterozygosity to CRISPR-Cas9 editing. A mixed MUT band comprising mostly HAND2^−/−^ senescent cells displayed a shorter bright MUT 260 bp band. Sanger sequencing of that MUT band confirmed deletions compared to the original sequence per the vector design (Appendix A).

## Data Availability

The data presented in this study are available on request from the corresponding author.

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
