# Peer review of "Distinct HAND2/HAND2-AS1 Expression Levels May Fine-Tune Mesenchymal and Epithelial Cell Plasticity of Human Mesenchymal Stem Cells"

_ijms, 2023, doi:10.3390/ijms242216546_

Round 1
Reviewer 1 Report
Comments and Suggestions for Authors
Journal: International Journal of Molecular Sciences
MS ID: ijms-2651859
Title: Distinct HAND2/HAND2-AS1 Expression Levels Finetune Mesenchymal and Epithelial Cell Plasticity in Human Mesenchymal Stem Cells.
This study was aimed to elucidate the enigma involving, HAND2-HAND2-AS1 regulation of MET in both tissue development homeostasis, and pathogenesis.
Major comments:
Overall, this paper is well written and looks scientific sound. However, there still many issues need major revision before acceptance.
Comments:
1. Please improve the resolution in each picture among of all figures ( Figure 1-Figure 5)。
2. Please include the label in each sub-data of either Figure 1, Figure 2 and Figure 3 or Figure caption.
3. Please provide the quantification data and sample number of VEGF, FGF-2 and EGF in the Figure 1.
4. Pease provide the quantification data and sample number in the Figure 2 (Adipocyte, chondrocyte and osteocytes)
5. Please provide the cell viability data of MSC in the Figure 3 after 2 to 4 weeks of incubation.
6. Please provide the quantification data of CD31 expression and the sample number in the Figure 3 either immunofluorescence staining assay and flow cytometry analysis.
7. Please enlarge the magnification and improve the resolution in the Figure 4(d).
8. Please provide the quantification data of colony formation in the Figure 5(b)。

Comments on the Quality of English Language
none
Reviewer 2 Report
Comments and Suggestions for Authors
The manuscript in question appears to be incomplete and exhibits several shortcomings. The authors observed a gene expression change and constructed a somewhat underdeveloped narrative, attributing their findings to serendipity. Here are the specific issues with the manuscript:
1. Lack of Experimental Rigor: This study lacks the essential components of a scientific investigation. The authors primarily focus on the creation of a CRISPR-KO cell line, failing to provide proper experimental validation of the clones. They rely on a rudimentary agarose gel (Figure 5), which is not only of poor quality but also lacks protein-level validation of the clones they have generated.
2. Inadequate Control and Presentation: The manuscript lacks proper control data, for e.g. Figure 3, how does the HAND2 protein level change from day 0 to different days? Additional, major shortcoming includes the absence of downstream analysis or further experimental work to substantiate the initial observation. The study appears to offer little to no scientific value. Due to the unvalidated clones, it does not even qualify as a resource paper.
3. Poor Writing and References: The manuscript's writing is subpar, with references marked but not properly inserted. This suggests a lack of thorough proofreading and attention to detail on the part of the authors.
In summary, the study design is fundamentally flawed, and it fails to make any significant scientific advancements. The data sets are poorly integrated, the bioinformatics analysis is minimal, and the agarose gel results are of low quality. Consequently, I must recommend the rejection of this manuscript based on these grounds.
Comments on the Quality of English Language
Must be improved
Round 2
Reviewer 1 Report
Comments and Suggestions for Authors
Authors were adequately responded all comments.
Comments on the Quality of English Language
none
Author Response
We sincerely thank the reviewer for the time, efforts, and insights provided, which helped improve the manuscript level!
The revised manuscript has now been English edited by a professional English editing service. Edits are highlighted in yellow in the now-revised manuscript. We hope you find it suitable for publication in the International Journal of Molecular Sciences.
Reviewer 2 Report
Comments and Suggestions for Authors
The authors have compelling reasons for the concerns I had. They have made some appropriate changes to manuscript. I accept the manuscript and suggest them to work on their writing to make this study better.
Comments on the Quality of English Language
English editing is needed!
Author Response

(The authors gave the same response as above.)
